# Analyzing the worldwide progression of COVID-19 cases and deaths using nonlinear mixed-effects model

Hiroki Koshimichi [1] *, Akihiro Hisaka [2]

1 Project Management Department, Shionogi & Co., Ltd., Osaka, Japan, 2 Graduate School of Pharmaceutical Sciences, Chiba University, Chiba, Japan

* hiroki.koshimichi@shionogi.co.jp

**Data Availability Statement:** All relevant data are within the paper and its Supporting information files.

## Abstract

COVID-19 is an infectious disease that continues to spread worldwide. A precise estimation of the cases and deaths due to COVID-19 would allow for appropriate consideration of healthcare resource allocation, public health response, and vaccination and economic planning, to minimize social damage. In this study, we analyzed the progression of COVID-19 cases and deaths until January 2022 in 156 countries using a nonlinear mixed-effect model based on the SIR framework. Given the major changes in mortality from infection, risk of reinfection and social responses, the analysis was limited to the period before the emergence of the Omicron variant. The impact of infection prevention measures in various countries was assessed, with a specific focus on estimating the effectiveness of lockdowns, where the effect was assumed to change over time. By accounting for excess mortality, our analysis allowed the estimation of unreported cases and deaths, and thus providing a more comprehensive understanding of the impact of pandemic. In the analysis, we identified gross domestic product (GDP), proportion of people aged 65 years or older, latitude of the capital city on transmissibility of infection, and city population and cardiovascular death rate on mortality rate as significant influencing factors. Furthermore, the differences in transmissibility and mortality rates by variants and the effect of vaccination on the mortality rate were assessed. The transmissibility has increased by odds ratios of 1.2 to 1.4 in Beta, Gamma, and Delta variants; mortality rate has increased by odds ratios of 1.7, 2.2, and 1.4 in Beta, Gamma, and Delta variants, respectively; and vaccination decreased the mortality rate by odds ratios of 0.4 and 0.1 in Delta and other variants, respectively.

## Introduction

COVID-19 is an infectious disease caused by SARS-CoV-2, which was first confirmed in Wuhan, China in December 2019, and has spread worldwide. Common symptoms include headache, loss of smell and taste, nasal congestion and rhinorrhea, cough, myalgia, sore throat, fever, diarrhea, and dyspnea. Most patients are asymptomatic or have mild symptoms; however some may become severe and fatal. Now the COVID-19 is no longer a "global health

**Funding:** The author(s) received no specific funding for this work.

**Competing interests:** The authors have declared that no competing interests exist.

emergency" [1], however, the virus is still circulating in the world and the risk remains of new variants emerging that cause new surges in cases and deaths, and other healthcare issues such as Long COVID remains [2]. Furthermore, information regarding infectious diseases other than COVID-19 and global health issues is being collected and made available [3].

The spread of infection changes due to various factors such as lockdown, travel restrictions, social distancing, wearing masks, hand-washing, polymerase chain reaction (PCR) tests and quarantine of patients, vaccination, and virus mutation. The degree and duration of the lockdown differed by country and city. Many countries have restricted entry from other countries or travel overseas to prevent the spread of COVID-19. Travel restrictions may be the most important factor at the beginning and end of a pandemic [4]. Social distancing such as staying at home, self-quarantine, or wearing masks was encouraged during the pandemic, and the spread of COVID-19 has been contained to some extent. Hand-washing is also effective in preventing infection at the individual level. PCR tests were performed to identify infected symptomatic patients and asymptomatic close contacts, which necessitates social distancing through quarantine to reduce contact and control the spread of infection.

Mutations of the virus may change the speed of infection spread, severity of symptoms, and mortality rate. Some variants of the SARS-CoV-2, including Alpha, Beta, Gamma, and Delta, were known as formerly circulating variants of concern (VOC) before 2022. These VOCs have higher transmissibility and virulence, and can decrease the effectiveness of public and social health measures, such as vaccines, with the potential to replace existing variants. In particular, the Alpha variant was first confirmed in the UK in September 2020, and approximately a quarter of cases in London were caused by a different strain by November [5], and around 60% by mid-December [6]. After the emergence of the Omicron variant in November 2021, this variant and its subtypes spread rapidly worldwide, attributing as the common cause of most cases.

A COVID-19 vaccine was developed and used worldwide in January 2021 to reduce the spread, severity, and mortality due to COVID-19. Many types of vaccines have been made available as of September 2023, including mRNA, adenovirus vector, inactivated virus, and subunit vaccines. Depending on the viral strain, the efficacies to prevent transmission of infection or lower the severity and mortality rate due to COVID-19 differ among vaccines [7], with their efficacies waning over time [8]. Some medications such as Nirmatrelvir, Remdesivir, Molnupiravir, are also available [9].

The mortality rate due to COVID-19 differs by period and country, and is influenced by the knowledge of COVID-19 infection, medical environment, as the oxygen shortage in hospitals might lead higher mortality rate at the beginning of pandemic [10], and patients background including age [11], viral strain, and vaccination. Moreover, to estimate the actual mortality rate, the proportion of deaths to all infected patients is difficult to obtain because many patients are asymptomatic and therefore have not been confirmed. Therefore, estimating the actual number of infected patients remains difficult, with some patients dying without confirmation of COVID-19, especially when healthcare capacities are overwhelmed. Some countries experienced this scenario when the infection spread uncontrollably and healthcare capabilities were overwhelmed [12, 13]. Cases of deaths related to unconfirmed COVID-19 can be estimated as "Excess death", where it is defined as the number of deaths from all causes during a period, and is beyond the expected deaths under normal conditions. To estimate the actual death rate due to COVID-19, the number of asymptomatic patients and excess deaths must be considered.

Several approaches have been reported to predict COVID-19 confirmed cases and deaths. An SIR model is a common and simple mathematical model to describe the time-profile of susceptible, infected, and recovered populations for infectious diseases [14]. Several SIR model-based expanded models have been reported, such as, considering the severity of the

infection, quarantine of infected patients, and lockdown [15], estimating the effect of non-pharmaceutical preventive measures [16], and assessing the impact of prior comorbidity on COVID-19 complications and COVID-19 reinfection [17]. In addition, some studies have used machine-learning approaches [18, 19]. Our previous study highlighted regional differences in transmissibility and death rates in the early phase of the pandemic [20]. Another study reported positive correlation between the number of cases in each country and population density [21]. Some demographic statistics in each country, such as population density and proportion of elderly people may contribute to the differences in transmissibility or mortality rate of COVID-19. However, quantitative estimation of the effect of those factors is difficult because the number of cases and deaths considerably changed within-country level, depending on several attributes, including period, measures to prevent infection in each country, viral mutation, and vaccination. The analysis of long-term data by eliminating the effect of measures to prevent infection in each country, viral mutation, and vaccination allows us to estimate country-specific parameters of COVID-19 infection including transmissibility or mortality rate, and can compare them with demographic statistics in each country.

Comparing the basic parameters of COVID-19 infection among countries from several studies is not appropriate owing to the differences in the model structures. In addition, a complex mathematical model requires many parameters to be estimated, and a data from a country may be inappropriate for precise parameter estimation. A nonlinear mixed-effects model is a popular approach for analyzing continuous individual data simultaneously, especially in the field of drug development [22]. It allows us to consider inter-individual differences and model the effect of covariates on model parameters. The analysis can increase the sample size by pooling the data from multiple individuals compared with the analysis of each individual, which leads to a more robust and reliable estimation of model parameters. Furthermore, the analysis considers the variability between individuals, which allows us to understand how the model parameters vary among individuals. For example, the clearance capacity of a drug is correlated with its body weight. Analysis using a nonlinear mixed-effects model allows for the estimation of the mean and variance of the model parameters, as well as the quantitative contribution of body weight to the clearance capacity.

Due to the COVID-19, humanity experienced for the first time a disease with a rapid infection rate and high mortality rate, with limited effective treatments where information on disease spread is shared simultaneously around the world. Based on this premise, the purpose of this study is to precisely assess how various factors affect the spread of infection. The data after the emergence of the omicron variant was significantly less reliable for information on the number of infections due to the low severity of symptoms and mortality rates. It was excluded from this study because it violates this premise. In this study, we constructed a mathematical model based on the SIR model to describe the time-profile of COVID-19 confirmed cases, confirmed deaths, and excess deaths worldwide. A nonlinear mixed-effects model approach was applied for the first time to analyze the time-profile of COVID-19 cases and deaths in the study; investigate the underlying inter-regional/inter-country difference in transmissibility of infection or death rate; as well as evaluate the effects of vaccination and differences among variants. This approach would be useful in the analysis of COVID-19 and other infectious diseases with fatal risks that spread worldwide.

## Materials and methods

### Dataset

COVID-19 confirmed cases and related confirmed deaths in each country as of January 2022 were downloaded from Our World in Data [23]. Daily cases and deaths were converted to

weekly cases and deaths, and were normalized by the population of the country to $10^8$. Countries with over $10^6$ population were included in this study; therefore, COVID-19 confirmed cases and deaths from 156 countries (50 from Africa, 24 from Americas, 44 from Asia, 35 from Europe, and 3 from Oceania) were included.

Excess deaths were estimated for countries with reported total deaths, COVID-19 confirmed deaths, and estimated COVID-19 unrelated deaths. The reported total deaths as of January 2022 were downloaded from the Human Mortality Database [24], except for those in Japan. Those for Japan as of January 2022 were downloaded from the website [25]. Excess deaths due to COVID-19 were estimated as follows; if the reported weekly total deaths exceeded the sum of the COVID-19 confirmed deaths and the upper limit of 95% confidence interval of the estimated COVID-19 unrelated deaths, then the excess deaths were estimated using Eq 1:

$$
\text{Excess deaths (weekly)} = \text{Reported total deaths} - \text{COVID-19 confirmed deaths} - \text{Estimated COVID-19 unrelated deaths} \tag{1}
$$

where COVID-19 unrelated deaths were estimated as the 5-year average of reported weekly deaths from 2015 to 2019, with the data downloaded from the Human Mortality Database. If the reported weekly total deaths were below these levels, the excess deaths were considered negligible and treated as below the limit of quantification (BLQ) in the analysis. Finally, the estimated excess deaths from 40 countries were included in this study.

The time-course of the proportion of each variant in each country, as of January 2022, was downloaded from the website [26]. If the proportion of each variant in a week was missing in a county, the missing value was filled in with the median value of the proportion of each variant in the week by region (Africa, Asia, the Americas, Europe, or Oceania).

The time-course of the proportion of vaccinated population in each country were also downloaded from Our World in Data. Differences in the type of vaccine were not considered in this study, therefore, the mean effect of vaccination, regardless of the type of vaccine, was assessed. It has been assumed that vaccination becomes effective immediately after the second dose of vaccination, and that the effect of vaccination declines gradually over time [8].

The duration of the lockdown in each country, state, and city, was downloaded from Wikipedia [27] as of January 2022 (S1 and S2 Tables). The effect of the lockdown was assumed to be the same if the lockdown was at the national, state, or city level. The average effect of lockdown in each country was assessed.

The beginning of infection in each country was defined as the point at which the weekly COVID-19 confirmed cases exceeded 10 and the normalized weekly COVID-19 confirmed cases exceeded 10 for four consecutive weeks. The cut-off week of the analysis was defined in each country where the proportion of Omicron variant exceeded 25%. The data after the emergence of Omicron variant was excluded from this analysis because of its lowered severity of the symptom and mortality rate, re-infection cannot be ruled out, but this model cannot be applied to re-infection cases.

## Model

A mathematical model based on the SIR model was constructed to describe the time-course of COVID-19 confirmed cases, confirmed deaths, and excess deaths in each country (Fig 1). In this model, it was assumed that all populations were susceptible; the quarantined population in each infected compartment lost the potential to infect others; each variant had a different potential for infection transmissibility and death rate; vaccination reduced the

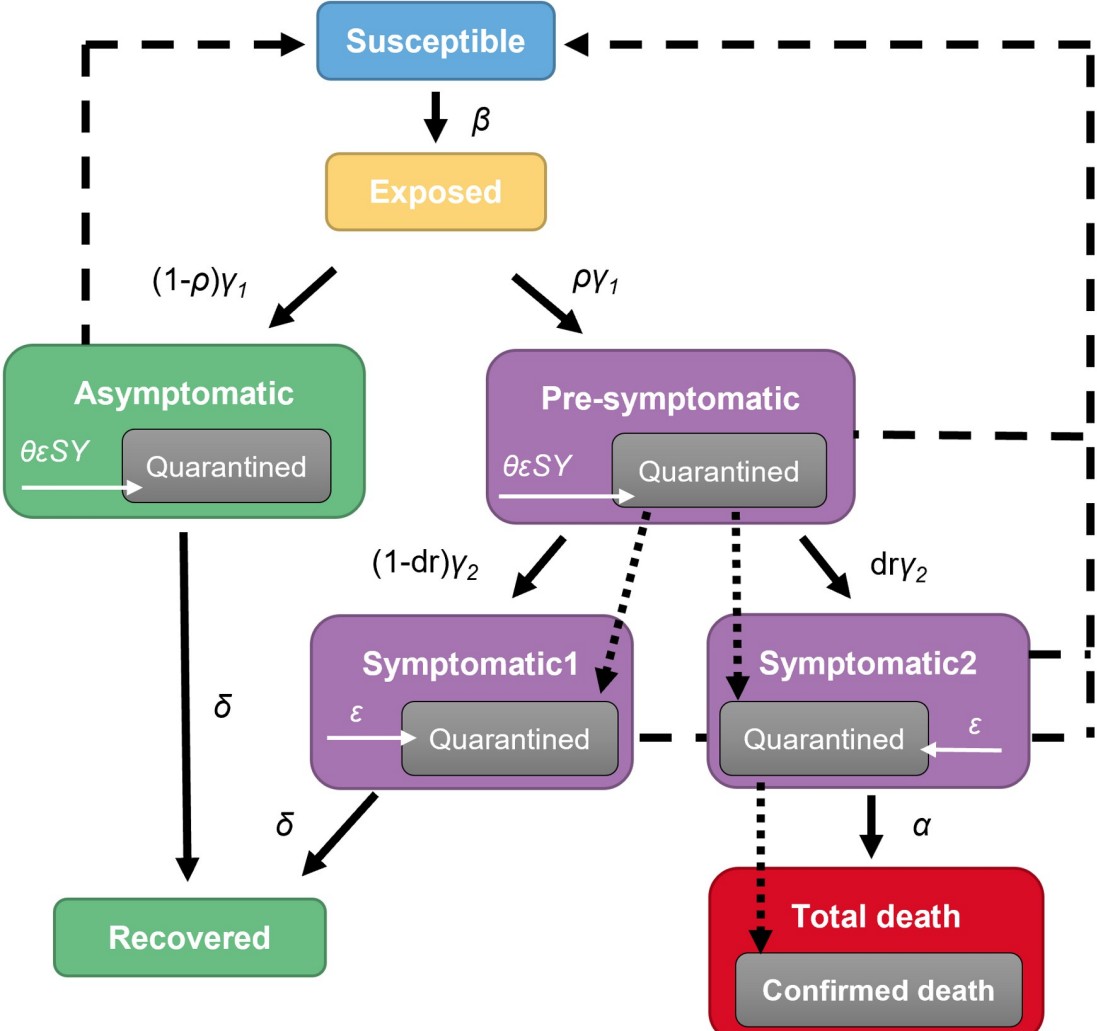

**Fig 1. Schematic of the model for transmission of SARS-Cov2 infection.** $\alpha$, $\beta$, $\gamma_1$, $\gamma_2$, $\delta$, $\varepsilon$, $\theta$ are rates of death, transmission, transfer from exposed to asymptomatic or pre-symptomatic, transfer from pre-symptomatic to symptomatic, recovery from asymptomatic or symptomatic, quarantine for symptomatic, and quarantine for pre-symptomatic or asymptomatic. Solid line shows transition from a compartment to the next compartment, dotted line shows transition from a compartment to the next compartment for quarantined population, and dashed line shows infection from symptomatic, pre-symptomatic, or asymptomatic population to susceptible population.

transmissibility of infection and death rate; and lockdown reduced the transmissibility of infection. Inter-country variability (ICV) was applied to the rate of transmission of infection ($\beta$), death rate of infected patients for wild type variant (dr), mean time from infection to death ($MT_{CDT}$), maximum rate of quarantine ($\varepsilon_{max}$), intercept of epsilon ($\varepsilon_{int}$), and maximum value of lockdown effect ($LD_{max}$). Inter-occasion variability (IOV) was applied to $\beta$ with estimating the parameters every four weeks. ICV was modeled using an exponential error model, except for dr'. ICV for dr' was modeled with additive error model. The estimated and fixed model parameters are listed in Table 1 and the density plots of the estimated parameters are shown in S1 Fig.

**Table 1. Estimated basic model parameters and covariates, effect of vaccination, and difference among viral variants.**

| Parameter (unit) | Description | Estimates (median [68% CI]) | Inter-country variability (%) |
|---|---|---|---|
| Basic model parameters | | | |
| $\beta$ (/cases/week*$10^{-9}$) | Rate of transmission of infection for wild type variant | 8.75 (8.52 to 8.99) | ICV: 22.6 (21.0 to 24.4) |
| | | | IOV: 62.8 (61.7 to 63.9) |
| dr (%) | Death rate of COVID-19 in the infected patients for wild type variant | 1.37 (1.29 to 1.46) | 74.3 (70.1 to 78.6) |
| $MT_{CDT}$ (week) | Mean time from infection to death (= $\ln2/\alpha$) | 1.16 (1.11 to 1.21) | 52.1 (48.6 to 56.0) |
| $\varepsilon_{max}$ (cases/week) | Maximum rate of quarantine for symptomatic population | 0.168 (0.146 to 0.193) | 153 (143 to 163) |
| $\varepsilon_{int}$ | Intercept of $\varepsilon$ | -0.0992 (-0.225 to 0.0153) | 115 (105 to 126) |
| $\varepsilon_{slp}$ | Slope of $\varepsilon$ | 0.0806 (0.0759 to 0.0855) | - |
| $LD_{max}$ | Maximum value of lockdown effect | 0.530 (0.440 to 0.607) | 226 (188 to 272) |
| $\lambda$ | Rate of decrease of lockdown effect | 0.0712 (0.0587 to 0.0813) | - |
| $E_0$ (cases) | Initial value of population in exposed compartment | 10.3 (9.35 to 11.3) | - |
| $Q_0$ | Initial value of quarantined proportion in each compartment | 0.0512 (0.0455 to 0.0576) | - |
| $Z_{Asy}$, $Z_{Pre}$ | Infectiousness of asymptomatic or pre-symptomatic versus symptomatic [28] | 0.35, 0.63 | |
| $\gamma1$, $\gamma2$ (1/week) | Mean half-life of time from infection to time of having infectiousness or time from having infectiousness to time of symptom onset [29] | 2.11, 1.94 | |
| $\delta$ (1/week) | Mean half-life of time from symptom onset to time of losing infectiousness [30] | 0.607 | |
| $\rho$ | Proportion of people who became symptomatic [28] | 0.80 | |
| $\theta$ (week/cases$^2$) | Rate of quarantine for asymptomatic or pre-symptomatic population [31] | $1.0*10^{-6}$ | |
| Covariate on model parameters | | | |
| Effect of logarithmic of GDP on $\beta$ | | 1.12 (0.904 to 1.34) | - |
| Effect of proportion of people aged 65 years or older on $\beta$ | | 0.141 (0.0979 to 0.185) | - |
| Effect of latitude of capital city on $\beta$ | | 0.0344 (0.0122 to 0.0543) | - |
| Effect of logarithmic of city population on the death rate | | 0.874 (0.486 to 1.27) | - |
| Effect of cardiovascular death rate on the death rate | | 0.560 (0.426 to 0.692) | - |
| Effect of vaccination | | | |
| $VacD_{Others}$ | Odds ratio of vaccination effect on the death rate for viral variants other than delta compared to non-vaccinated population | 0.105 (0.082 to 0.134) | - |
| $VacD_{Delta}$ | Odds ratio of vaccination effect on the death rate for Delta variant compared to non-vaccinated population | 0.364 (0.339 to 0.391) | - |
| Difference by virus variant | | | |
| $Var_{Alpha}$ | Odds ratio of transmissibility of infection for Alpha variant to wild type variant | 0.822 (0.776 to 0.873) | - |
| $Var_{Beta}$ | Odds ratio of transmissibility of infection for Beta variant to wild type variant | 1.35 (1.18 to 1.57) | - |
| $Var_{Gamma}$ | Odds ratio of transmissibility of infection for Gamma variant to wild type variant | 1.23 (1.07 to 1.40) | - |
| $Var_{Delta}$ | Odds ratio of transmissibility of infection for Delta variant to wild type variant | 1.27 (1.24 to 1.31) | - |
| $VarD_{Alpha}$ | Odds ratio of death rate for Alpha variant to wild type variant | 0.884 (0.848 to 0.922) | - |
| $VarD_{Beta}$ | Odds ratio of death rate for Beta variant to wild type variant | 1.72 (1.54 to 1.93) | - |
| $VarD_{Gamma}$ | Odds ratio of death rate for Gamma variant to wild type variant | 2.20 (1.98 to 2.45) | - |
| $VarD_{Delta}$ | Odds ratio of death rate for Delta variant to wild type variant | 1.38 (1.34 to 1.42) | - |
| Proportional residual error | | | |
| $\sigma_1$ | Proportional residual error of weekly confirmed cases | 0.204 (0.202 to 0.206) | - |
| $\sigma_2$ | Proportional residual error of weekly confirmed deaths | 0.261 (0.259 to 0.263) | - |
| $\sigma_3$ | Proportional residual error of weekly excess deaths | 0.0264 (0.0259 to 0.0268) | - |

CI = confidence interval, ICV = Inter-country variability, IOV = Inter-occasion variability

The differential equations of the mathematical model are shown in Eqs 2–11.

$$\frac{dS}{dt} = -\beta'' S\left(SY1' + SY2' + Z_{Asy}Asy' + Z_{Pre}Pre'\right) \tag{2}$$

$$\frac{dE}{dt} = \beta'' S\left(SY1' + SY2' + Z_{Asy}Asy' + Z_{Pre}Pre'\right) - \gamma_1 E \tag{3}$$

$$\frac{dAsy}{dt} = (1 - \rho)\gamma_1 E - \delta Asy \tag{4}$$

$$\frac{dQ_{Asy}}{dt} = -\delta Q_{Asy} + \theta\varepsilon(SY1' + SY2')Q_{Asy} \tag{5}$$

$$\frac{dPre}{dt} = \rho\gamma_1 E - \gamma_2 Pre \tag{6}$$

$$\frac{dQ_{Pre}}{dt} = -\gamma_2 Q_{Pre} + \theta\varepsilon(SY1' + SY2')Q_{Pre} \tag{7}$$

$$\frac{dSY1}{dt} = (1 - dr')\gamma_2 Pre - \delta SY1 \tag{8}$$

$$\frac{dQ_{SY1}}{dt} = (1 - dr')\gamma_2 Q_{Pre} + \varepsilon SY1' - \delta Q_{SY1} \tag{9}$$

$$\frac{dSY2}{dt} = dr''\gamma_2 Pre - \alpha Q_{SY2} - \alpha SY2' \tag{10}$$

$$\frac{dQ_{SY2}}{dt} = dr''\gamma_2 Q_{Pre} + \varepsilon SY2' - \alpha Q_{SY2} \tag{11}$$

where S is susceptible, E is exposed, Asy is asymptomatic, Pre is pre-symptomatic, SY1 is symptomatic 1, and SY2 is symptomatic 2 compartment cases. Qx is quarantined cases in each compartment of x. Apostrophe (') in each compartment means unquarantined cases in the compartment. The effect of variants, vaccination, and lockdown are modeled in β' and dr' as follows:

$$\beta'_i = \beta \times \exp(\eta_i + \kappa_{i,t}) \times Var_{i,t,variant} \times Vac_{i,t} \times LD_{i,t} \tag{12}$$

$$dr'_{logit,i} = dr_{logit} + \eta_i + \log(VarD_{i,t,variant}) + \log(VacD_{i,t,variant}) \tag{13}$$

dr' was modeled using a logit model (Eq 13) and finally converted to the actual death rate. The differences of transmissibility of infection by variants were estimated as the ratio to wild type variant; effect of vaccination on β and its decrease was fixed with the reported value [8]; and differences of death rate by variants and the effect on death rate by vaccination were estimated as odds ratio to wild type variant or unvaccinated population. The lockdown effect was modeled as follows, assuming that the maximum effect of lockdown ($LD_{max,i,t}$) decreases linearly by time and becomes negligible after $T_{last}$, the effect of lockdown ($LD_{i,t}$) is the same with $LD_{max,i,t}$ during the lockdown period, and $LD_{i,t}$ decreases exponentially by time after the end

of lockdown period.

$$LD_{max,i,t} = LD_{max,i} + \left(1 - LD_{max,i}\right) \times time/T_{last} \qquad (14)$$

$$LD_{i,t} = 1 - LD_{max,i,t} \qquad (15)$$

$$LD_{i,t} = 1 - LD_{max,i,t} \times \exp\left(-\lambda \times \left(time - T_{ref}\right)\right) \qquad (16)$$

where time is a week from the beginning of infection, $\lambda$ is a rate of decrease of lockdown effect, and $T_{ref}$ is a week from the latest timing of the end of lockdown. $T_{last}$ was estimated manually to minimize the objective function (OBJ), where 13, 26, 39, 52, 65, 78, and 91 weeks were tested, and 78 weeks (1.5 years) showed the lowest OBJ.

## Covariate model

Covariate model was tested to explore factors which can explain inter-regional/inter-country differences of $\beta$ or dr after the base model was constructed. The tested factors (candidate covariates) are listed in S3 Table. Covariate modeling was performed in three steps. First, we selected a combination of factors to avoid multicollinearity. Combinations of the candidate of covariates with r-squared values < 0.7 were selected. After selection, principal component analysis (PCA) was performed for each combination and the components with < 0.8 of cumulative proportion of variance were remained. Second, multiple regression analysis was performed with the estimated inter-country variability of $\beta$ or dr in the base model and the remained components. The factors tested in the nonlinear mixed-effect model analysis in the final step were selected if the r-squared value was the smallest among the combinations and if the factors were included in the remaining components in the PCA. Last, all factors were included in the model, and a nonlinear mixed-effect model analysis was performed (Eqs 17 and 18). If several factors were not statistically significant (p<0.05), they were removed from the model, and the analysis was performed again. Finally, a covariate model was constructed with, all covariates being statistically significant (p<0.05).

$$\beta''_i = \beta'_i \times \prod_j \left( p_j \times \frac{Cov_{j,i} - median\left(Cov_j\right)}{median\left(Cov_j\right)} \right) \qquad (17)$$

$$dr''_{logit,i} = dr'_{logit,i} + \sum_j \left( q_j \times \frac{Cov_{j,i} - median\left(Cov_j\right)}{median\left(Cov_j\right)} \right) \qquad (18)$$

where $Cov_j$ is the j-th covariate, and $p_j$ and $q_j$ are the effect of the j-th covariate on $\beta$ and dr, respectively.

## Analysis

NONMEM (ver 7.4, ICON plc) was used for the analysis with a nonlinear mixed-effect model. The Markov chain Monte Carlo method was used for parameter estimation with a burn-in of 10 thousand, iteration 30 thousand, and thinning every 10 iterations in the analysis [32]. The M3 method [33] was applied to the BLQ data, in which less than one confirmed cases or one confirmed death was treated as the BLQ. The BLQ for excess death has been described in

Dataset section. The NONMEM code is available on Github at https://github.com/h-koshi/WorldwideCovid19.

## Results

The time-course of COVID-19 infected cases and, deaths until January 2022 in 156 countries was analyzed using a nonlinear mixed-effect model that expanded from the SIR model. The model successfully described the time-course of these data at the global, regional (Figs 2 and 3), and national levels (Figs 4–6, S2–S4 Figs). Moreover, the excess deaths due to COVID-19 were satisfactorily described by this model, where the data were available (Fig 6) for countries with negligible (Japan and Australia) or non-negligible (UK, US, Italy, and France) excess deaths. The overall mean value of β was estimated as 8.75 x $10^{-9}$/cases/week with inter-country variability of 22.6% and inter-occasion variability of 62.8%, indicating that the transmissibility of infection changes immensely from time to time and the changes were greater than inter-country differences. The overall mean death rate was estimated at 1.37% for wild type viral variant, which was remarkably higher than that of influenza (~0.01% [34]).

The estimated transmissibility of the infection (β) and death rate for each country are shown in Fig 7. β was relatively high in the Americas and Europe region, and low in Africa. Conversely, the death rate was relatively high in Africa and South America but low in Europe. Those in Asia and Oceania region were middle.

The effect of lockdown on β was estimated and its time-course and maximum effect are shown in Fig 8. The estimated lockdown profile peaked from March to June 2020, during the first wave of the pandemic. The lockdown effect drastically decreased after the end of each

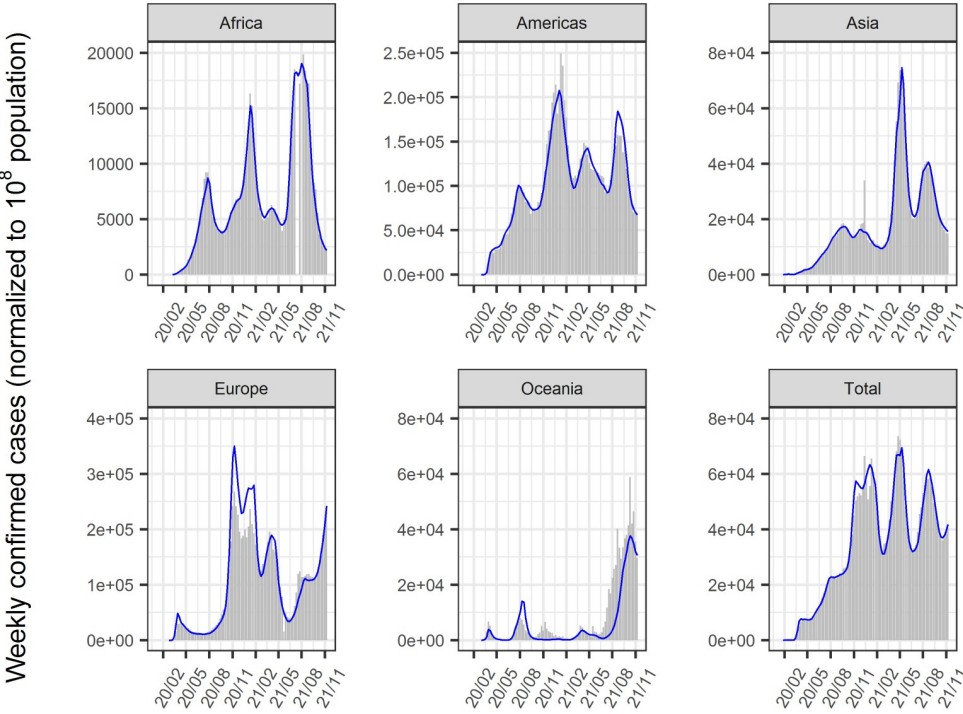

**Fig 2. Reported and predicted weekly confirmed cases globally and each region.** Gray bar: reported weekly confirmed cases. Blue line: predicted weekly confirmed cases. The population in each region was normalized to $10^8$ in the analysis.

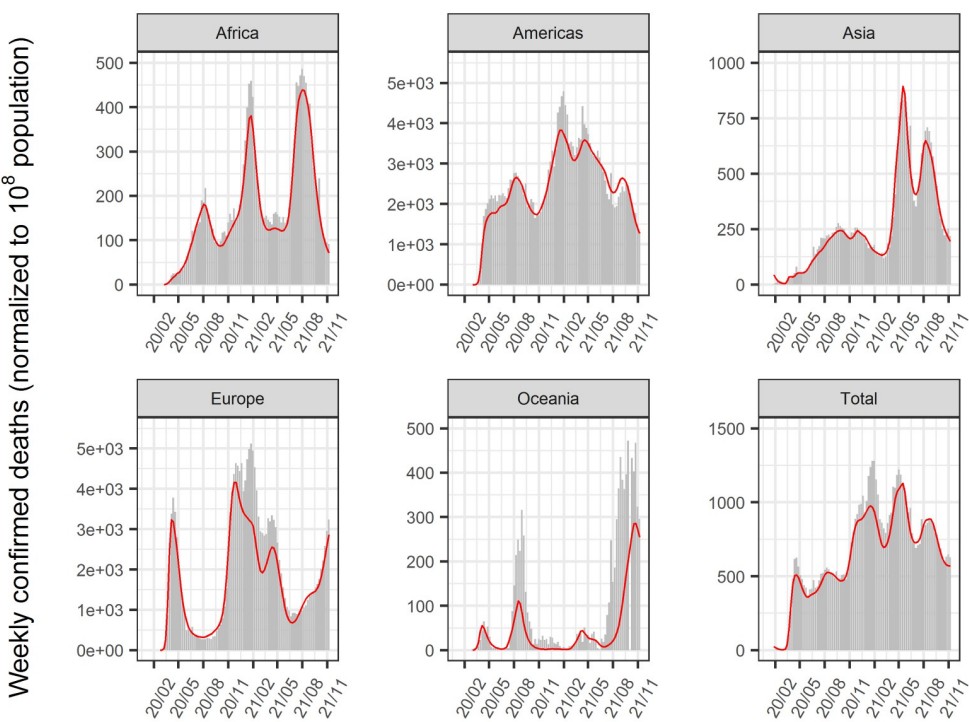

**Fig 3. Reported and predicted weekly confirmed deaths globally and each region.** Gray bar: reported weekly confirmed deaths. Red line: predicted weekly confirmed deaths. The population in each region was normalized to $10^8$ in the analysis.

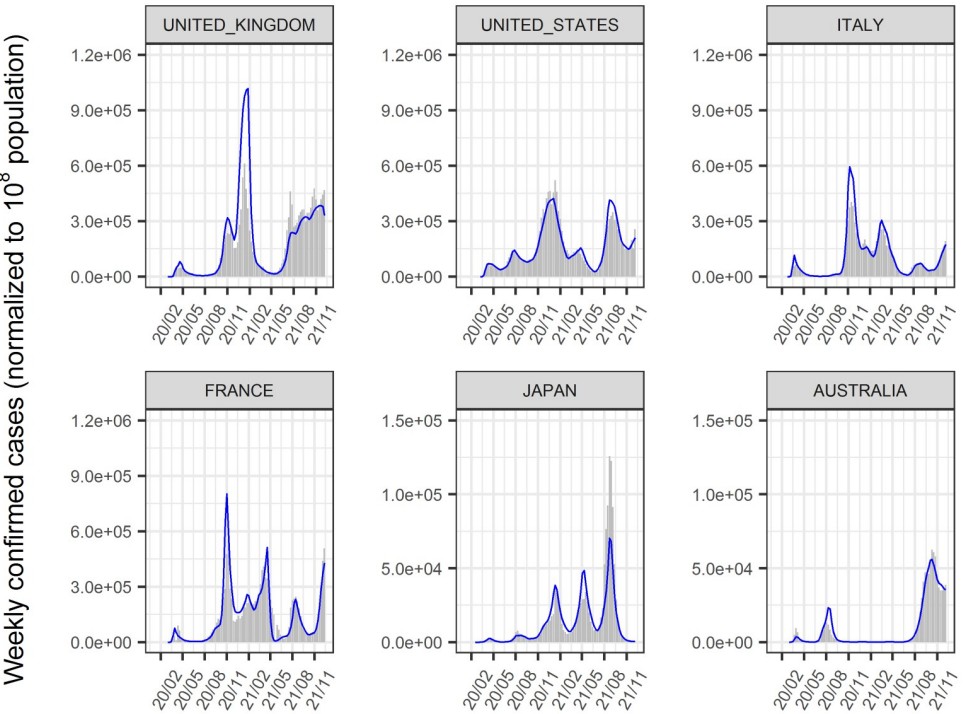

**Fig 4. Reported and predicted weekly confirmed cases.** Gray bar: reported weekly confirmed cases. Blue line: predicted weekly confirmed cases. The population in each country was normalized to $10^8$ in the analysis. Representative countries are shown. S2 Fig shows the results for all countries.

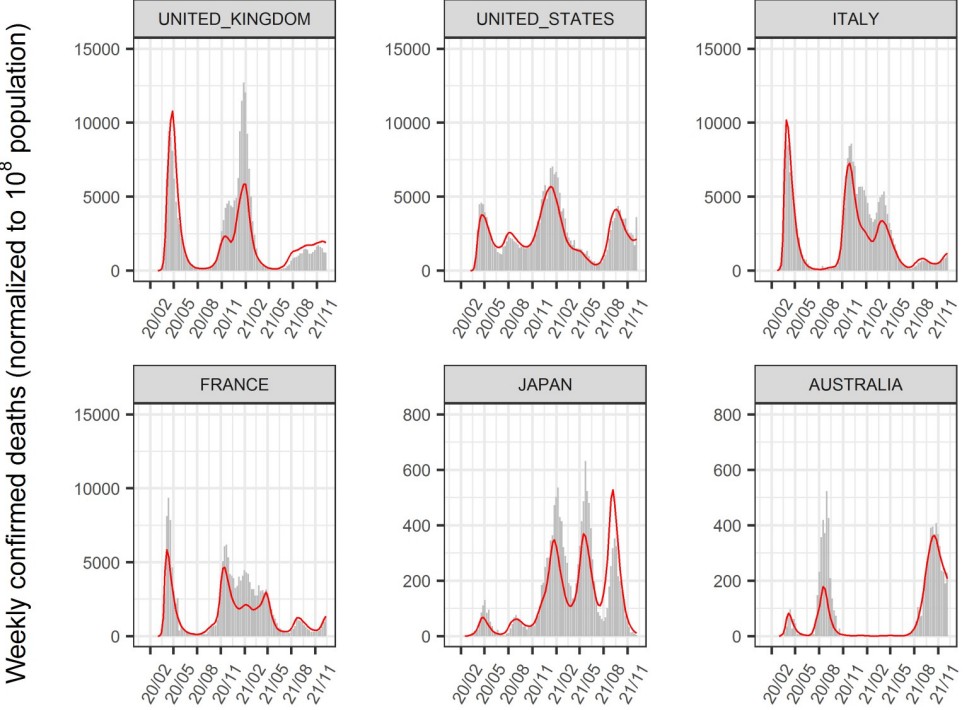

**Fig 5. Reported and predicted weekly confirmed deaths.** Gray bar: reported weekly confirmed deaths. Red line: predicted weekly confirmed deaths. The population in each country was normalized to $10^8$ in the analysis. Representative countries are shown. S3 Fig shows the results for all countries.

lockdown period, suggesting that human behavior and activities quickly returned to the pre-lockdown levels soon after the end of the lockdown. The maximum lockdown effect was relatively high in Europe and low in Africa, suggesting that the lockdown is highly effective in countries where the basic transmissibility of infection is high, as in Europe.

The estimated time-course of the inter-occasion variability of β is shown in Fig 9. The estimates for each country change greatly throughout the duration; however, the mean values are approximately zero throughout the period and are not biased, suggesting that the major causes of within-country β difference were appropriately modeled and successfully estimated in this study.

Five covariates were identified as the basic characteristics that could explain inter-region/inter-country differences of transmissibility or death rate. The logarithmic of GDP, proportion of people aged 65 years or older, and latitude of the capital city were found to positively affect the transmissibility of COVID-19, indicating that economically active countries have higher transmissibility and that the older adult populations are more susceptible to the infection. Logarithmic of city population and cardiovascular death rate were found to positively affect the death rate due to COVID-19, which is consistent with a report that cardiovascular disease has four-fold higher mortality rates in COVID-19 patients [35].

Differences in transmissibility and death rates for each variant were assessed (Fig 10). The estimated odds ratios were lower than one for Alpha variant, suggesting that the variant had lower transmissibility and death rates than those in the wild-type variant. In contrast, the Beta, Gamma, and Delta variants showed odds ratios higher than one, suggesting that these variants had higher transmissibility and death rates than the wild-type variant. These estimates are

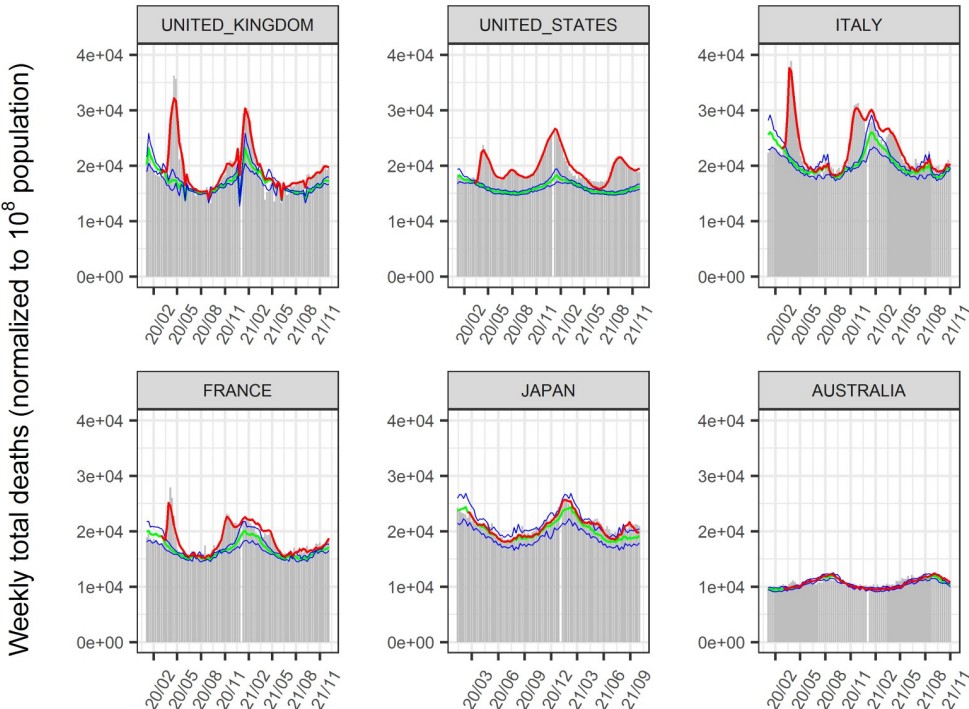

**Fig 6. Reported and predicted weekly total deaths.** Total death is a sum of COVID-19 related confirmed death, COVID-19 related unconfirmed death, and death from other reasons. Gray bar: reported weekly total deaths. Red line: predicted weekly total deaths. Green line and blue line: mean and its 90% confidence intervals of expected death from other reasons. The population in each country was normalized to $10^8$ in the analysis. Representative countries are shown. S4 Fig shows the results for all countries.

similar to those estimated in another report [36] with increased transmissibility of 29%, 25%, 38%, and 97% for Alpha, Beta, Gamma, and Delta variants, respectively, indicating that the difference in the transmissibility of infection or death rate can be estimated with this population approach using a mathematical model.

The effect of vaccination on the death rate of patients with COVID-19 was estimated by variants (Fig 11). The estimated odds ratios were lower than one for all variants, suggesting that vaccination was effective in lowering the risk of death from COVID-19. However, the estimated vaccination effect was weaker in the delta variant than in the other variants, suggesting that the effectiveness of the vaccination differ among variants and that the vaccine needs to be updated in response to prevalent strains.

## Discussion

The number of COVID-19 cases and deaths have rapidly changed owing to the behavioral change of people towards the infection, preventive measures in each country, presence of variants, and vaccination status. The prediction of cases and deaths due to COVID-19 is important from the several viewpoints, including economics, medical resource preparation, and preventive measures, and several prediction methods and results have been reported. The two main approaches can be categorized as model-based analysis and machine learning techniques, such as artificial intelligence (AI). AI analysis has some advantages, such that, it can recognize high-level patterns and relationships without human bias. However, the analysis is influenced by

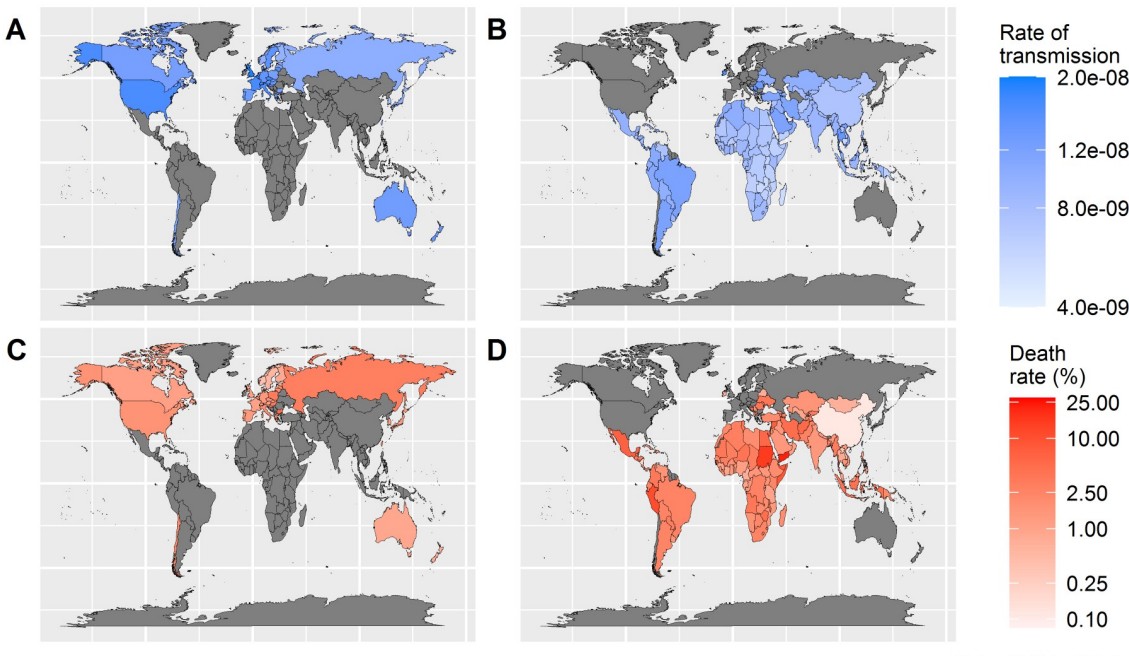

**Fig 7. World maps of the estimated rate of transmission (β) and death rate.** A: β for countries with excess death data. B: β for countries without excess death data. C: death rate for countries with excess death data. D: death rate for countries without excess death data.

data quality and quantity, and requires considerable computational resources, and often operates as a black box. Conversely, model-based analysis defines some meaningful and visible parameters, which can be interpreted, and the reliability can be assessed. Furthermore, the prediction of future cases and deaths is easier in model-based analysis under the assumption that the parameters, such as transmissibility of the infection or mortality rate, are changed for some reasons, including, the emergence of new variants, vaccination, and usage of medicines. In this study, the time-course of COVID-19 confirmed cases, confirmed deaths, and excess deaths were successfully described with model-based approach using the proposed SIR-based nonlinear mixed-effect model global, regional, and national levels. This model considers the inter-occasion difference of transmissibility of the infection, lockdown effect, difference in the transmissibility and death rate by variants, and vaccination effect on the transmissibility and death rate.

The transmissibility and death rates differ among regions/countries [20, 37]. The reasons for this difference remain unclear, and several factors affect them in a complicated manner and change with time. Therefore, an analysis that focuses on one or more countries, or short-term data analysis, is not appropriate to clarify the reason for the difference. The simultaneous analysis of the infection status in many countries worldwide in the long-term would be useful for discussing the cause of inter-regional/inter-country differences. In this study, an analysis with nonlinear mixed-effect model was performed and some factors, which can explain the inter-regional/inter-country difference in β or dr were identified. The candidates for the covariate were first screened by PCA and then the effect was estimated in the analysis using a nonlinear mixed-effect model in order to avoid multicollinearity. Demographic data, such as population, population density, and the proportion of people aged 65 years or older, and economic data, such as GDP, were selected as candidates for the covariates in this study. However,

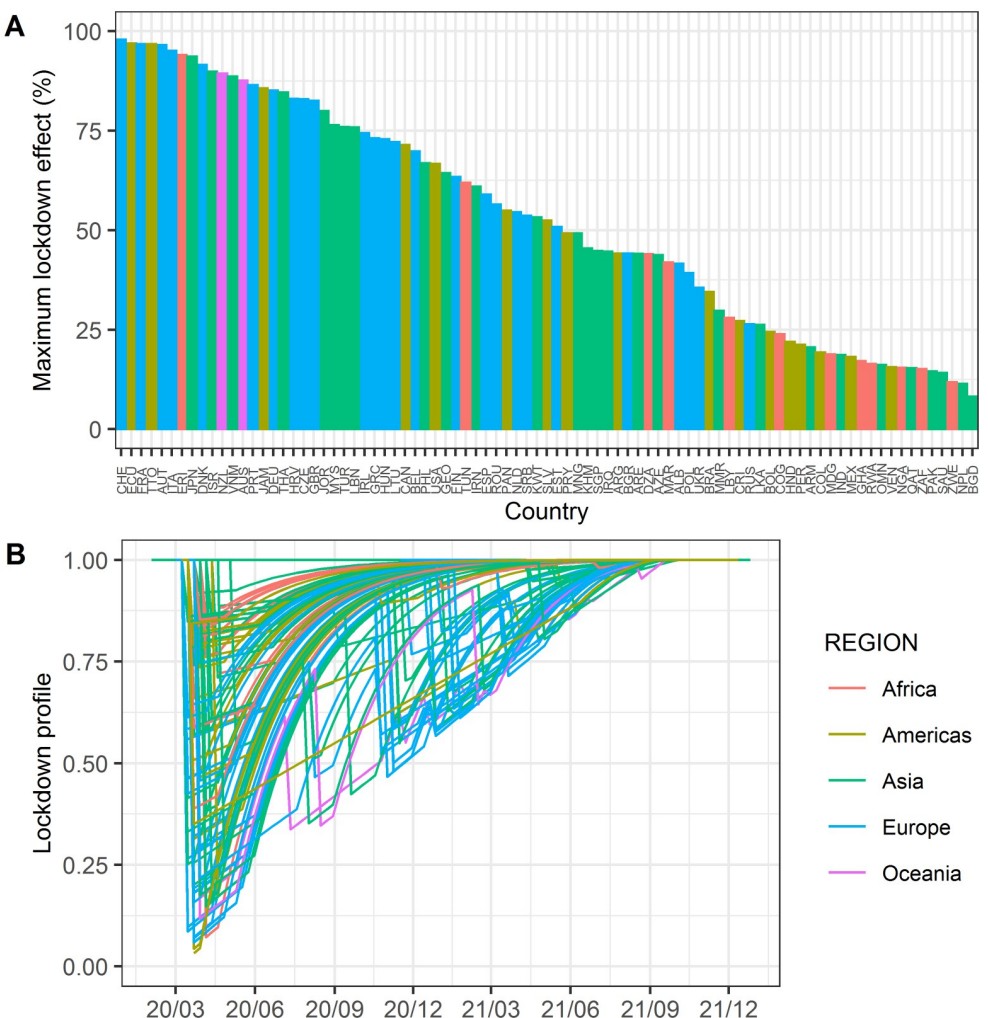

**Fig 8. Estimated lockdown effect.** A: Estimated maximum lockdown effect (%) by country. B: Estimated lockdown effect profile by country.

these data are known to have some correlation with each other. PCA facilitated in effectively identifying the covariates. This analysis showed that GDP, proportion of people aged 65 years or older, and latitude of the capital city are positively correlated with the transmissibility of the infection (β), suggesting that countries with high GDP and older adult population in the Northern hemisphere are at high susceptible to the easy transmission of infections. Thus, preventive measures, such as lockdowns, are most effective in countries with these characteristics. Furthermore, this analysis showed that the city population and cardiovascular death rate were positively correlated with death rate due to COVID-19. Early hospitalization may reduce the risk of COVID-19-related mortality in countries with these features. This study successfully identified covariates that efficiently explained transmissibility or death rates from commonly used and easily accessible indicators. Many economic, medical, population and geographic features of the nation are not independent index and correlated complexly with numerous characteristics. As a matter of fact, among the many correlated features, it is impossible and thus unimportant to identify which one has the particularly greatest impact on the spread of infection. Therefore, in this study, many country-specific characteristics were dimensionally

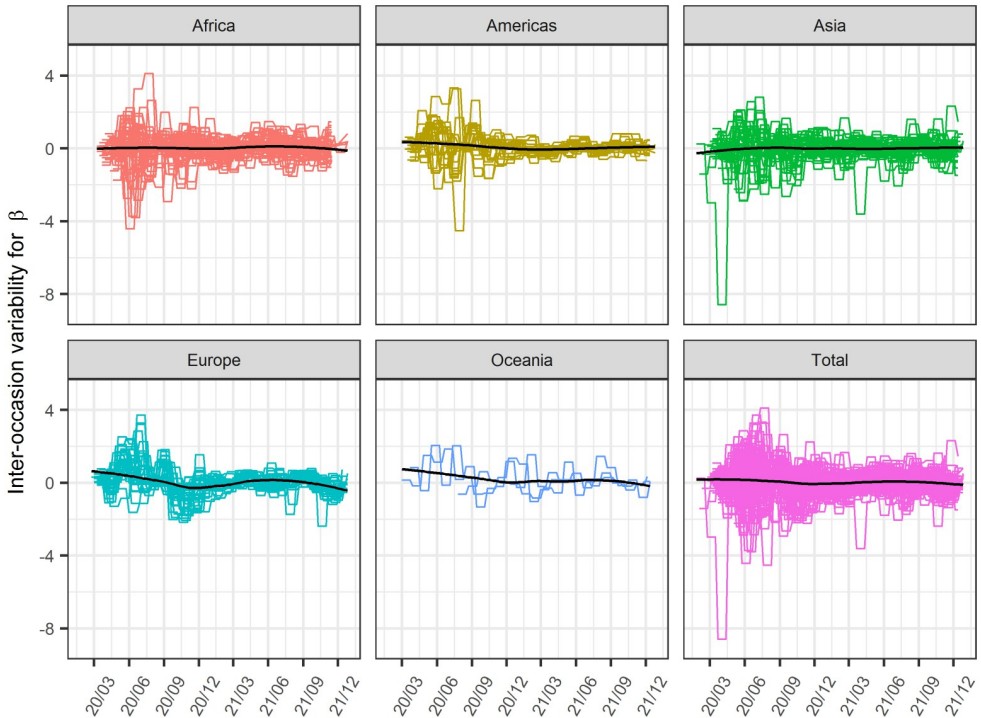

**Fig 9. Time-course of estimated inter-occasion variability for β in each region.** Black line: locally estimated scatterplot smoothing line.

compressed by the principal component analysis and analyzed impact of their independent component on the infection rate, and then a model was constructed to explain changes in the infection rate by limiting the analysis to the important factors among the detected components. As a result, we obtained interesting results showing that factors such as GDP and the

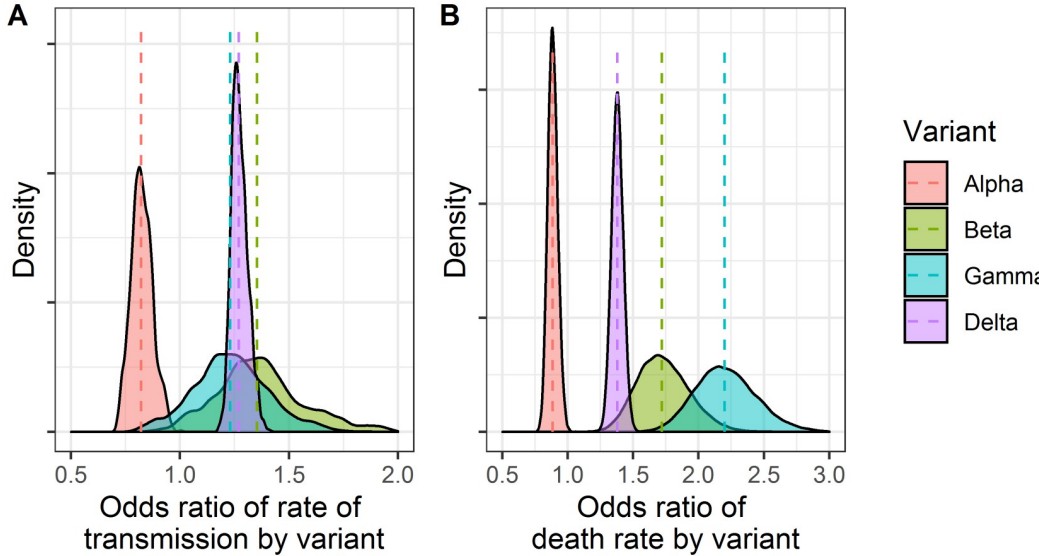

**Fig 10. Density plots of estimated rate of transmission and death rate by variant.** A: estimated odds ratio of rate of transmission by variant to wild-type variant. B: estimated odds ratio of death rate by variant to wild-type variant.

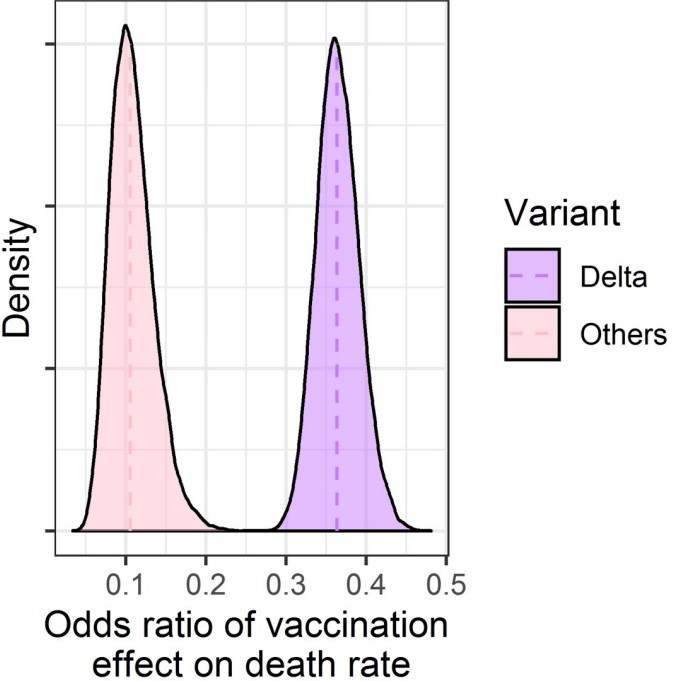

**Fig 11. Density plots of estimated odds ratio of vaccination effect on death rate.**

proportion of aged people 65 years or older affect the infection rate. It should be noted, however, that it would be going too far to consider these factors as independent and distinct influencing factors considering the methodology of this study, and that they are representative of one of the similar characteristics of the trends and may have indirect effects and possible confounding.

Mixed-effect model analysis enables the separation of fixed effects and random effects and identifies significant covariates that influence fixed effects. It assumes that each country is relatively independent and has a similar informational significance. Many of these conditions cannot be applied to countries of different sizes and environments. However, the extent of the spread of infection varies greatly from country to country, which reflecting the differences in the levels of human traffic and the strong influence of policies and culture on infection in each country. Therefore, adopting a mixed-effects model in this study is reasonable, which would have allowed us to identify the various factors that influence infection. The transmissibility of the infection has changed owing to preventive measures against the infection in each country, and lockdown is one of the most effective means. In this study, effects by the various preventive measures of infection including lockdown were firstly evaluated as temporal inter-occasion variability. However, the estimated inter-occasion variability was high especially in the first year of the pandemic and gradually decreased to low levels, suggesting rapid changes in their effectiveness especially in the initial period. Therefore, we assumed that the lockdown effect should be estimated separately from the effect of other preventive measures. As a matter of fact, the lockdown effect was explained appropriately by this model and evaluated as negligible after approximately a year from the begging of the pandemic. It was suggested that the lockdown effect became negligible after 1.5 years from the beginning of COVID-19 pandemic. The effects of the lockdown were reasonably estimated in this study. This study does not take into consideration the extent of lockdown measures, specifically the differences between

national-level lockdowns and city-level lockdowns. Lockdowns are highly effective in preventing the spread of infections; however, they impose significant burdens on society, necessitating their implementation at the appropriate timing and level. Therefore, conducting a more detailed analysis of the duration and extent of lockdowns is considered useful in maximizing the efficiency of lockdown in future potential pandemic. The social responses other than lockdown cannot be modeled in this study because it was difficult to collect the detailed quantitative information of social responses from worldwide countries, and they were assumed to be included in the random temporal inter-occasion variability.

Fig 9 shows that the inter-occasion variability was not biased throughout the entire period. However, a slight region-dependent tendency was observed in inter-occasion variability. In this study, data from each country were assumed to be independent and unrelated to each other; however, the transmission of infection across countries cannot be ignored especially in neighboring countries. In this case, a similar time-course of inter-occasion variability is expected in neighboring countries. Inter-occasion variability may be reduced and made more stable when traffic across countries and inter-country transmission of infection are considered precisely in the model.

In this study, the temporal and regional variability in the effectiveness of infection prevention measures were treated as a random inter-occasion variability, and their causal relationship is not captured. Fig 9 shows that during the first half year of the pandemic, large inter-occasion variability is estimated compared to the other periods, suggesting that there may have been significant differences in the temporal and regional variability in the effectiveness of infection prevention measures during this period.

Estimating the difference in the transmissibility of infection or death rate by variants only from local observations proves difficult, and the reported estimates differ largely among studies [36, 38] as the regions, populations, and/or periods are different in each study. This study did not limit the regions, populations, or periods; therefore, the estimated difference in the transmissibility of infection or death rate by variants is expected to be close to the true difference among variants. Furthermore, the estimated odds ratios may be different from the other reports since this study considers unreported cases and deaths in the calculation of odds ratios. It is reported that the probability of hospitalization is high in Delta variant compared to the other variants [39], suggesting that the symptoms of COVID-19 are severe in patients infected with Delta variant. The difference of the estimated effect of vaccination on mortality rate between Delta variant and other variants may result from the difference in the severity of disease. The vaccination should be recommended as public health strategy especially when a high transmissibility or a high severity variant emerges. The estimated difference in transmissibility for Beta or Gamma variants was less reliable; that is, the distribution of the sampling was wider than those of the Alpha or Delta variants. This may a result of the lack of data on Beta or Gamma variants, since these variants spread only in certain regions or countries, unlike Alpha or Delta variants, which spread globally.

The effect of vaccination was assessed in some clinical studies, which reported the effect on transmissibility, admission rate, disease severity to some degree. However, the effect on the death rate was difficult to precisely assess because the death rate was originally low (1%), and the vaccination would lower it; therefore, the number of participants required to assess the vaccination effect on death rate becomes unrealistically high in clinical studies. In such cases, the modeling approach would be useful for estimating the effect because the model analysis can use worldwide data, not limited to specific region/country and period, and long-term analysis makes the comparison between data on pre-, during, and post-vaccination possible. This analysis of vaccine effectiveness is unique in that it integrates results from countries all over the world with very different characteristics, and quantitatively takes into account differences

among countries in terms of lockdown, economic size, and the effects of viral mutations, which may have led to a more objective evaluation. On the other hand, the type and kind of vaccine could not be differentiated in this study because of the lack of data. Furthermore, the use of medication other than vaccine was not considered in this study, which may affect the estimation of the effect of vaccination. For the purpose of testing the model parameter sensitivity, we have simulated the COVID-19 cases and deaths if there were no vaccines available in the world as supplementary data in S5 Fig. Based on the simulation, due to vaccination, the COVID-19 weekly cases and deaths would be decreased by approximately 70% by November 2021, about a year after the first vaccine became available. This suggests that the early development of vaccines and their proactive administration are extremely important and had saved many lives.

In this study, excess deaths enabled estimation of the actual mortality rate due to COVID-19. However, unlike confirmed cases and deaths (156 countries), excess death information was limited to 40 countries; therefore, the estimated model parameters for each country may be less reliable or biased for countries without excess death information than those with it. Furthermore, most of the countries whose excess deaths being available were Europe and developed countries, therefore, the bias in the distribution of excess death data in a particular country may lead to bias in the estimation of some model parameters. The predictive accuracy is expected to be improved by the inclusion of abundant and precise estimations of excess death data when infection is fatal and medical collapse occurs.

The Omicron variant and other variants after the emergence of Omicron were excluded from this study because its characteristics, such as transmissibility of infection and mortality rate, were relatively different from those of the other variants. Because of the lowered severity and mortality rate of the Omicron variant, preventive measures in each country have been eased, human behavior has gradually returned to pre-pandemic patterns, and the frequency of proactive PCR testing has significantly decreased. Therefore, the quantity of the confirmed cases and deaths became less reliable. This study aimed to analyze infected cases and deaths with relatively high mortality rates by accounting for excess deaths, and thus the Omicron variant and its subtypes were considered unsuitable for inclusion in the analysis. Furthermore, owing to the high transmissibility of the Omicron variant, the number of cases increased drastically after its emergence, and neglecting patients who were repeatedly infected became increasingly challenging. This model does not consider the repeated infection, therefore, when examining patients with repeated infections, for example, the pandemic last for a long time enough to lose individual immunity against the virus or a variant emerges which have characteristics to escape the immunity, it is crucial to consider a model that accounts for the possibility of recovered individuals losing their immunity and rejoining susceptible populations.

## Conclusion

In this study, we analyzed the time-course of COVID-19 cases and deaths in countries worldwide using SIR-based mathematical model and a nonlinear mixed-effects model. The actual death rate was estimated by considering excess deaths due to COVID-19. We identified some factors that can explain the inter-country differences in the transmissibility of the infection or COVID-19-related death rate. Furthermore, the analysis allowed us to estimate the effects of preventive measures for lockdowns, differences among variants, and effects of vaccination. This analysis showed that lockdown is a quite effective preventive measures of the transmission of infection in the beginning of the pandemic, however, the effect of lockdown decreases over time and reached to a negligible level after a certain period of time. The reason the lockdown quickly lost its effectiveness may be because society could not bear the strain, suggesting

that combining various kind of more acceptable preventive measures are important, such as vaccination and medications. Therefore, development of vaccines and medicines are required as soon as possible after the spread of infectious disease in the world. Furthermore, to estimate the true mortality rate under the pandemic like COVID-19, to build a global system for demographic trend that allows us to understand excess deaths is also important. The approach proposed in this paper would be helpful in understanding and characterizing the worldwide transmission of infection and how to effectively prevent the transmission of infection for COVID-19 and the other potential future outbreaks of infectious diseases.

## Supporting information

**S1 Fig. Density plots of the estimated model parameters.**
(PDF)

**S2 Fig. Reported and predicted weekly confirmed cases in each country.**
(PDF)

**S3 Fig. Reported and predicted weekly confirmed deaths in each country.**
(PDF)

**S4 Fig. Reported and predicted weekly total deaths in each country.**
(PDF)

**S5 Fig. Simulated weekly cases and deaths if there were no vaccines available in the world.**
(PDF)

**S1 Table. Lockdown start and end date in each country, territory, and place.**
(DOCX)

**S2 Table. Lockdown start and end date in Japan in each place.**
(DOCX)

**S3 Table. The tested factors list in the covariate modeling.**
(DOCX)

## Acknowledgments

The authors would like to thank H Sato and H Yoshioka for their valuable comments.

## Author Contributions

**Conceptualization:** Hiroki Koshimichi.

**Data curation:** Hiroki Koshimichi.

**Formal analysis:** Hiroki Koshimichi.

**Funding acquisition:** Hiroki Koshimichi, Akihiro Hisaka.

**Investigation:** Hiroki Koshimichi.

**Methodology:** Hiroki Koshimichi, Akihiro Hisaka.

**Project administration:** Hiroki Koshimichi.

**Resources:** Hiroki Koshimichi.

**Software:** Hiroki Koshimichi.

**Supervision:** Hiroki Koshimichi, Akihiro Hisaka.

**Validation:** Hiroki Koshimichi.

**Visualization:** Hiroki Koshimichi.

**Writing – original draft:** Hiroki Koshimichi.

**Writing – review & editing:** Akihiro Hisaka.

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
