## [Decision Letter · Decision Letter 0]

8 Mar 2024

PONE-D-23-37811Analyzing the worldwide progression of COVID-19 cases and deaths using nonlinear mixed-effects modelPLOS ONE

Dear Dr. Koshimichi,

Thank you for submitting your manuscript to PLOS ONE. After careful consideration, we feel that it has merit but does not fully meet PLOS ONE’s publication criteria as it currently stands. Therefore, we invite you to submit a revised version of the manuscript that addresses the points raised during the review process.

We look forward to receiving your revised manuscript.

Kind regards,

Mohammad Nayeem Hasan

Academic Editor

PLOS ONE

Journal Requirements:

3. We note that Figure 7 in your submission contain map images which may be copyrighted. All PLOS content is published under the Creative Commons Attribution License (CC BY 4.0), which means that the manuscript, images, and Supporting Information files will be freely available online, and any third party is permitted to access, download, copy, distribute, and use these materials in any way, even commercially, with proper attribution. For these reasons, we cannot publish previously copyrighted maps or satellite images created using proprietary data, such as Google software (Google Maps, Street View, and Earth). For more information, see our copyright guidelines: http://journals.plos.org/plosone/s/licenses-and-copyright.

(1) You may seek permission from the original copyright holder of Figure 7 to publish the content specifically under the CC BY 4.0 license.  

Reviewers' comments:

Reviewer's Responses to Questions

**Comments to the Author**

1. Is the manuscript technically sound, and do the data support the conclusions?

Reviewer #1: Yes

Reviewer #2: Yes

2. Has the statistical analysis been performed appropriately and rigorously? 

Reviewer #1: Yes

Reviewer #2: Yes

3. Have the authors made all data underlying the findings in their manuscript fully available?

Reviewer #1: Yes

Reviewer #2: Yes

4. Is the manuscript presented in an intelligible fashion and written in standard English?

Reviewer #1: Yes

Reviewer #2: Yes

5. Review Comments to the Author

Reviewer #1: I have read this paper. The authors have analyzed the progression of COVID-19 cases and deaths until January 2022 in 156 countries using a nonlinear mixed-effect model based on the SIR framework. Given the major changes in mortality from infection, risk of re-infection and social responses, the analysis was limited to the period before the emergence of the Omicron variant. The impact of infection prevention measures in

various countries was assessed, with a specific focus on estimating the effectiveness of lockdowns, where the effect was assumed to change over time. By accounting for excess mortality, our analysis allowed the estimation of unreported cases and deaths, and thus providing a more comprehensive understanding of the impact of pandemic. In the analysis, we identified gross domestic product (GDP), proportion of people aged 65 years or older, latitude of the capital city on transmissibility of infection, and city population and cardiovascular death rate on mortality rate as significant influencing factors. Furthermore, the differences in transmissibility and mortality rates by variants and the effect of vaccination on the mortality rate were assessed. The transmissibility has increased by odds ratios of 1.2 to 1.4 in Beta, Gamma, and Delta variants; mortality rate has increased by odds ratios of 1.7, 2.2, and 1.4 in Beta, Gamma, and Delta variants, respectively; and vaccination decreased the mortality rate by odds ratios of 0.4 and 0.1 in Delta and other variants, respectively.

The idea of the manuscript is interesting and it can be considered further for a place of publication in Plos One. However, I haver the following queries for the authors:

1. Can authors provide more details on the specific nonlinear mixed-effect model based on the SIR framework that you employed?

2. What led to the decision to limit the analysis to the period before the emergence of the Omicron variant, and how might this choice influence the generalizability of your findings?

3. Elaborate on how the assumption of changing effectiveness of lockdowns over time was incorporated into the model and its implications on the study outcomes.

4. Could you explain in more detail how excess mortality was accounted for in your analysis and how it contributed to estimating unreported cases and deaths?

5. How were social responses, such as public compliance with preventive measures, integrated into the model, and how might these factors impact the validity of your results?

6. Provide further insights into how GDP, proportion of people aged 65 or older, latitude, city population, and cardiovascular death rate were identified as significant influencing factors on mortality rates.

7. Can you elaborate on how the odds ratios for transmissibility and mortality rates were derived for Beta, Gamma, and Delta variants, and how these results align with existing literature?

8. Clarify the methodology used to assess the impact of vaccination on mortality rates, and discuss any potential confounding variables that might influence these results.

9. Provide additional context on the relatively high odds ratio for mortality in the Delta variant compared to other variants, and discuss the potential implications for public health strategies.

10. Detail the sources of data used in your analysis, including information on the reliability and representativeness of the datasets for the countries considered.

11. Can you conduct sensitivity analyses or validation studies to assess the robustness of your statistical methods, especially considering the complexity of the model?

12. Explicitly discuss the assumptions made in the model and their potential impact on the results, addressing any limitations or uncertainties associated with these assumptions.

13. Explore the temporal and regional variability in the effectiveness of infection prevention measures and discuss how these variations might influence your overall findings.

14. Address potential biases in your study, such as reporting biases or selection biases, and discuss their implications for the interpretation of your results.

15. In the conclusion, provide a more detailed discussion on the practical implications of your findings for policymakers and public health practitioners, including recommendations for future interventions.

16. Check the suggested papers: https://doi.org/10.1016/j.chaos.2020.110032;
https://doi.org/10.1002/oca.2748.

Reviewer #2: The findings of the research have the potential to get published. However, the manuscript in its current form needs minor revision. Following suggestions should be incorporated while preparing the revised version of the manuscript.

1. It is suggested to write a paragraph on death due to oxygen shortage in hospitals, in the introduction.

2. Is there any source that provides the data of another health issue post-COVID-19?

6. PLOS authors have the option to publish the peer review history of their article (what does this mean?). If published, this will include your full peer review and any attached files.

Reviewer #1: **Yes: **Andrew Omame

Reviewer #2: No

---

## [Author Response · Author response to Decision Letter 0]

8 May 2024

May 8, 2024

Dr. Mohammad Nayeem Hasan

Editor-in-Chief

PLOS ONE

Dear Dr. Mohammad Nayeem Hasan,

Thank you for your instructions and forwarding the reviewer's thoughtful feedbacks regarding our manuscript, analyzing the worldwide progression of COVID-19 cases and deaths using nonlinear mixed-effects model. We agree with you that our paper needs some revision to meet PLOS ONE’s publication criteria, and we have amended this by changing the content. The amended contents are explained from next page. We also responded to all queries made by the reviewers and modified the manuscript appropriately.

With these changes to our final manuscript, we hereby resubmit our manuscript for a secondary evaluation. Thank you once again for your consideration of our paper.

Sincerely yours,

Hiroki Koshimichi, Ph.D.

Clinical Pharmacology & Pharmacokinetics,

Project Management Department, 

Shionogi & Co., Ltd., 

8F, Nissay Yodoyabashi East Bldg., 3-13, Imabashi 3-chome, 

Chuo-ku, Osaka, 541-0042, Japan

Phone: +81-70-7812-6705

e-mail: hiroki.koshimichi@shionogi.co.jp

Journal Requirements:

>1

Thank you for your comment. I have followed PLOS ONE's style request and uploaded files with appropriate naming.

>2

I have uploaded model code file of the analysis to Github with open-source license. The link to the Github repository is added in manuscript Materials and Methods, Analysis section. (https://github.com/h-koshi/WorldwideCovid19)

3. We note that Figure 7 in your submission contain map images which may be copyrighted. All PLOS content is published under the Creative Commons Attribution License (CC BY 4.0), which means that the manuscript, images, and Supporting Information files will be freely available online, and any third party is permitted to access, download, copy, distribute, and use these materials in any way, even commercially, with proper attribution. For these reasons, we cannot publish previously copyrighted maps or satellite images created using proprietary data, such as Google software (Google Maps, Street View, and Earth). For more information, see our copyright guidelines: http://journals.plos.org/plosone/s/licenses-and-copyright.

(1) You may seek permission from the original copyright holder of Figure 7 to publish the content specifically under the CC BY 4.0 license. 

>3

The map of Figure 7 was created with R package “maps v3.4.1”, in which the vector data of the world map was imported from public domain Natural Earth data project (the 1:50m world map, version 2.0, the latest version available in 2015).

I have revised Figure 7 to include the data source, as “Made with Natural Earth.” following the instruction https://www.naturalearthdata.com/about/terms-of-use/. 

 

Reviewer #1

Dear Dr. Andrew Omame,

Thank you for providing your thoughtful insights. We have revised our manuscript based on your fruitful comments as below.

1. Can authors provide more details on the specific nonlinear mixed-effect model based on the SIR framework that you employed?

>1

In accord with the reviewer’s comment, I have uploaded the NONMEM model code of the analysis as supporting information to provide the detail of the model, which is now available on Github at https://github.com/h-koshi/WorldwideCovid19. I mentioned in the Method that the code is posted on Github.

2. What led to the decision to limit the analysis to the period before the emergence of the Omicron variant, and how might this choice influence the generalizability of your findings?

>2

Considering the reviewer's query, the below sentence was added in the Introduction section to clarify our objective in this study. (page 4, line 10)

"Due to the COVID-19, humanity experienced for the first time a disease with a rapid infection rate and high mortality rate, with limited effective treatments where information on disease spread is shared simultaneously around the world. Based on this premise, the purpose of this study is to precisely assess how various factors affect the spread of infection. The data after the emergence of the omicron variant was significantly less reliable for information on the number of infections due to the low severity of symptoms and mortality rates. It was excluded from this study because it violates this premise. “

The reasons for our choice to exclude the Omicron variant have also been explained in the discussion section, that is, lowered severity of the symptom and mortality rate, less reliable reported cases and deaths after the emergence of Omicron variant, and repeated infection cannot become negligible. Our study aimed to analyze infected cases and deaths with relatively high mortality rates by accounting for excess deaths, and thus the Omicron variant and its subtypes were considered unsuitable for inclusion in the analysis. 

The below sentence was added in the materials and methods section for clarification. (page 5, line 29)

“The data after the emergence of Omicron variant was excluded from this analysis because of its lowered severity of the symptom and mortality rate, re-infection cannot be ruled out, but this model cannot be applied to re-infection cases.” 

3. Elaborate on how the assumption of changing effectiveness of lockdowns over time was incorporated into the model and its implications on the study outcomes.

>3

Considering the reviewer's comment, the below sentences were added in the discussion section. (page 23, lines 6 and 15)

“In this study, effects by the various preventive measures of infection including lockdown were firstly evaluated as temporal inter-occasion variability. However, the estimated inter-occasion variability was high especially in the first year of the pandemic and gradually decreased to low levels, suggesting rapid changes in their effectiveness especially in the initial period. Therefore, we assumed that the lockdown effect should be estimated separately from the effect of other preventive measures. As a matter of fact, the lockdown effect was explained appropriately by this model and evaluated as negligible after approximately a year from the begging of the pandemic.”

“This study does not take into consideration the extent of lockdown measures, specifically the differences between national-level lockdowns and city-level lockdowns. Lockdowns are highly effective in preventing the spread of infections; however, they impose significant burdens on society, necessitating their implementation at the appropriate timing and level. Therefore, conducting a more detailed analysis of the duration and extent of lockdowns is considered useful in maximizing the efficiency of lockdown in future potential pandemic.”

4. Could you explain in more detail how excess mortality was accounted for in your analysis and how it contributed to estimating unreported cases and deaths?

>4

The method for accounting the excess mortality has been described in the Materials and Methods (Dataset), especially by Equation 1. Furthermore, I have uploaded the NONMEM model code of the analysis for more detailed understanding of how excess mortality was incorporated in this model and how it contributed to estimating unreported cases and deaths. 

Significance of the excess death is described in the Discussion section as follows. (page 24, line 36)

“In this study, excess deaths enabled estimation of the actual mortality rate due to COVID-19. However, unlike confirmed cases and deaths (156 countries), excess death information was limited to 40 countries; therefore, the estimated model parameters for each country may be less reliable or biased for countries without excess death information than those with it. Furthermore, most of the countries whose excess deaths being available were Europe and developed countries, therefore, the bias in the distribution of excess death data in a particular country may lead to bias in the estimation of some model parameters. The predictive accuracy is expected to be improved by the inclusion of abundant and precise estimations of excess death data when infection is fatal and medical collapse occurs.” 

It is difficult to state the extent to which the excess deaths affected the accuracy of the analysis, since there are large differences among countries, but it is possible that there was a significant impact in countries with serious infection situations, and that the accuracy of the analysis might be limited in some countries where there was no information on excess deaths.

5. How were social responses, such as public compliance with preventive measures, integrated into the model, and how might these factors impact the validity of your results?

>5

As the reviewer suggested, public compliance with preventive measures possibly affect the infection spread. However, it varies extensively country by country and thus it was very difficult to collect the information of detailed quantitative social responses from worldwide countries. Therefore, the social responses other than lockdown effect were modeled as random temporal inter-occasion variability in this model. Considering the reviewer's suggestion, the below sentence was added in the discussion section. (page 23, line 21)

“The social responses other than lockdown cannot be modeled in this study because it was difficult to collect the detailed quantitative information of social responses from worldwide countries, and they were assumed to be included in the random temporal inter-occasion variability.”

6. Provide further insights into how GDP, proportion of people aged 65 or older, latitude, city population, and cardiovascular death rate were identified as significant influencing factors on mortality rates.

>6

To respond the reviewer's query, the below sentence was modified in the discussion section. (page 22, line 21)

“Notably, they may include direct or indirect factors, and it is not possible to deny confounding between covariates. A careful interpretation is necessary.”

to

“Many economic, medical, population and geographic features of the nation are not independent index and correlated complexly with numerous characteristics. As a matter of fact, among the many correlated features, it is impossible and thus unimportant to identify which one has the particularly greatest impact on the spread of infection. Therefore, in this study, many country-specific characteristics were dimensionally compressed by the principal component analysis and analyzed impact of their independent component on the infection rate, and then a model was constructed to explain changes in the infection rate by limiting the analysis to the important factors among the detected components. As a result, we obtained interesting results showing that factors such as GDP and the proportion of aged people 65 years or older affect the infection rate. It should be noted, however, that it would be going too far to consider these factors as independent and distinct influencing factors considering the methodology of this study, and that they are representative of one of the similar characteristics of the trends and may have indirect effects and possible confounding.”

7. Can you elaborate on how the odds ratios for transmissibility and mortality rates were derived for Beta, Gamma, and Delta variants, and how these results align with existing literature?

>7

I have uploaded the NONMEM model code of the analysis to support understanding of how the odds ratios are estimated. Furthermore, the below sentence was added in the result and discussion section. (page 19, line 16, and page 24, line 5)

“with increased transmissibility of 29%, 25%, 38%, and 97% for Alpha, Beta, Gamma, and Delta variants, respectively”

“Furthermore, the estimated odds ratios may be different from the other reports since this study considers unreported cases and deaths in the calculation of odds ratios.”

8. Clarify the methodology used to assess the impact of vaccination on mortality rates, and discuss any potential confounding variables that might influence these results.

>8

I have uploaded the NONMEM model code of the analysis to clarify the methodology to assess the impact of vaccination on mortality rate. The below sentence was added in the discussion section. (page 24, line 24)

“This analysis of vaccine effectiveness is unique in that it integrates results from countries all over the world with very different characteristics, and quantitatively takes into account differences among countries in terms of lockdown, economic size, and the effects of viral mutations, which may have led to a more objective evaluation. On the other hand, the type and kind of vaccine could not be differentiated in this study because of the lack of data. Furthermore, the use of medication other than vaccine was not considered in this study, which may affect the estimation of the effect of vaccination.”

9. Provide additional context on the relatively high odds rat

---

## [Decision Letter · Decision Letter 1]

26 Jun 2024

Analyzing the worldwide progression of COVID-19 cases and deaths using nonlinear mixed-effects model

PONE-D-23-37811R1

Dear Dr. Hiroki Koshimichi,

We’re pleased to inform you that your manuscript has been judged scientifically suitable for publication and will be formally accepted for publication once it meets all outstanding technical requirements.

Kind regards,

Mohammad Nayeem Hasan

Academic Editor

PLOS ONE

Additional Editor Comments (optional):

Comments from PLOS Editorial Office: We note that one or more reviewers has recommended that you cite specific previously published works in an earlier round of revision. As always, we recommend that you please review and evaluate the requested works to determine whether they are relevant and should be cited. It is not a requirement to cite these works and you may remove them before the manuscript proceeds to publication. We appreciate your attention to this request.

Reviewers' comments:

Reviewer's Responses to Questions

**Comments to the Author**

1. If the authors have adequately addressed your comments raised in a previous round of review and you feel that this manuscript is now acceptable for publication, you may indicate that here to bypass the “Comments to the Author” section, enter your conflict of interest statement in the “Confidential to Editor” section, and submit your "Accept" recommendation.

Reviewer #1: All comments have been addressed

Reviewer #2: All comments have been addressed

2. Is the manuscript technically sound, and do the data support the conclusions?

Reviewer #1: Yes

Reviewer #2: Yes

3. Has the statistical analysis been performed appropriately and rigorously? 

Reviewer #1: Yes

Reviewer #2: Yes

4. Have the authors made all data underlying the findings in their manuscript fully available?

Reviewer #1: Yes

Reviewer #2: Yes

5. Is the manuscript presented in an intelligible fashion and written in standard English?

Reviewer #1: Yes

Reviewer #2: Yes

6. Review Comments to the Author

Reviewer #1: All my queries have been addressed and the paper can now be accepted for publication in this journal.

Reviewer #2: (No Response)

7. PLOS authors have the option to publish the peer review history of their article (what does this mean?). If published, this will include your full peer review and any attached files.

Reviewer #1: No

Reviewer #2: No

---

## [Editor Report · Acceptance letter]

2 Jul 2024

PONE-D-23-37811R1 

PLOS ONE

Dear Dr. Koshimichi, 

I'm pleased to inform you that your manuscript has been deemed suitable for publication in PLOS ONE. Congratulations! Your manuscript is now being handed over to our production team.

Kind regards, 

on behalf of

Dr. Mohammad Nayeem Hasan 

Academic Editor

PLOS ONE